# Non-invasive Optical Technical Identification of Red Pigments on Chinese Paper Notes

**Jie Ren** [1], **Cunjin Gao** [1], **Jigang Wang** [1,*], **Yang Shen** [1], **Jilong Shi** [1], **Quanxiao Liu** [1,*] and **Wei Chen** [2,*]

[1] Beijing Key Laboratory of Printing and Packaging Materials and Technology, Beijing Institute of Graphic Communication, Beijing 102600, China; 18811127099@163.com (J.R.); gcj2019015220@163.com (C.G.); SY18811127900@163.com (Y.S.); jilongshi@bigc.edu.cn (J.S.)

[2] Department of Physics, The University of Texas at Arlington, Arlington, TX 76019-0059, USA

[*] Correspondence: jigangwang@bigc.edu.cn (J.W.); drllqx@163.com (Q.L.); weichen@uta.edu (W.C.)

**Abstract:** Red pigments with bright colors were widely used in ancient Chinese painted pottery, books, antiques, calligraphy, and paintings. Herein, red pigments of traditional paper notes were investigated by non-invasive optical technology in order to enrich the Chinese historical pigments knowledge base. The results of laser Raman spectroscopy tests on five paper notes clearly identified the inorganic mineral pigments including ocher and cinnabar. Infrared spectroscopy measurements indicated that an artificial synthetic magenta was employed as the organic pigment. Inorganic and organic red pigments were applied together on the same samples 2 and 5 which can be speculated to serve an anti-counterfeiting function. In addition, SEM-EDS analysis of sample 5 clearly showed that the red pigment was composed of lead oxides and ZnS was added as color modulator. Combined with the abovementioned non-invasive techniques, analysis of printed pigments can provide a feasible method to authenticate and conserve paper notes.

**Keywords:** Chinese paper notes; red pigments; non-invasive identification; Raman spectroscopy; optical technology

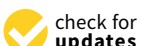



## 1. Introduction

Red pigments were some of the most commonly used pigments in ancient China due to their bright colors. For example, the inorganic red pigments on painted pottery at the Yangshao cultural site and on a Northern Wei Dynasty pottery figurine were demonstrated to be $Fe_2O_3$ [1]. A red HgS lacquerware pigment was found at the Zhejiang Yuyao Hemudu site and in the Jiangsu Xuyi Han tomb [2]. The red pigments HgS and $Pb_3O_4$ were widely applied in Dunhuang murals [3–6]. Numerous red pigments in the ancient building materials of the Summer Palace of Beijing included $Fe_2O_3$, $Pb_3O_4$, and synthetic organic red pigments [7–9]. Red pigments are also widely used in modern Chinese bills and stamps, such as the business licenses of the Qing Dynasty and the Republic of China, and the imperial China engraved coiling dragon 2-point and 4-point stamps [10]. During our research on the material of stamps, we found that there are many reports on the use of pigments abroad, such as the Hawaiian missionary stamps issued by the Hawaiian Islands in 1851 [11], Mauritius stamps in 1847 [12], stamps and banknotes in the region of Rijeka in Croatia in 1918 [13], and a set of stamps in the Italian Kingdom and Republic [14]. In addition, red pigments are widely used in painting, printing, dyeing, sealing style, and other fields.

The Chinese Printing Cultural Heritage Research Center has collected a large amount of paper notes. The research methods of scientific and technological history workers for cultural relic collections generally involve site visits and investigations, followed by observation with the eyes, hands and other traditional methods. However, these traditional identification methods sometimes have some shortcomings, and the use of scientific and technological means can make up for these shortcomings and increase the accuracy of

identification. Herein, research on pigments can help identify the time that the paper notes were fabricated and suggest methods for preserving them. Considering the non-regenerative and valuable characteristics of paper notes, it is necessary to apply in situ non-invasive analytical techniques.

## 2. Experimental Samples and Instrumentation

### 2.1. Experimental Samples

All samples were collected by Chinese Printing Cultural Heritage Research Center, Beijing Institute of Graphic Communication (BIGC). Figure 1a shows the five stamps (labeled from Sample 1 to Sample 5) analyzed in this study set. Figure 1b shows the images obtained by optical microscopy, taking the points on the pictures of sample 3 and sample 5 as examples, the points on the two samples are photographed through an optical microscope, and the magnification used is 10×.

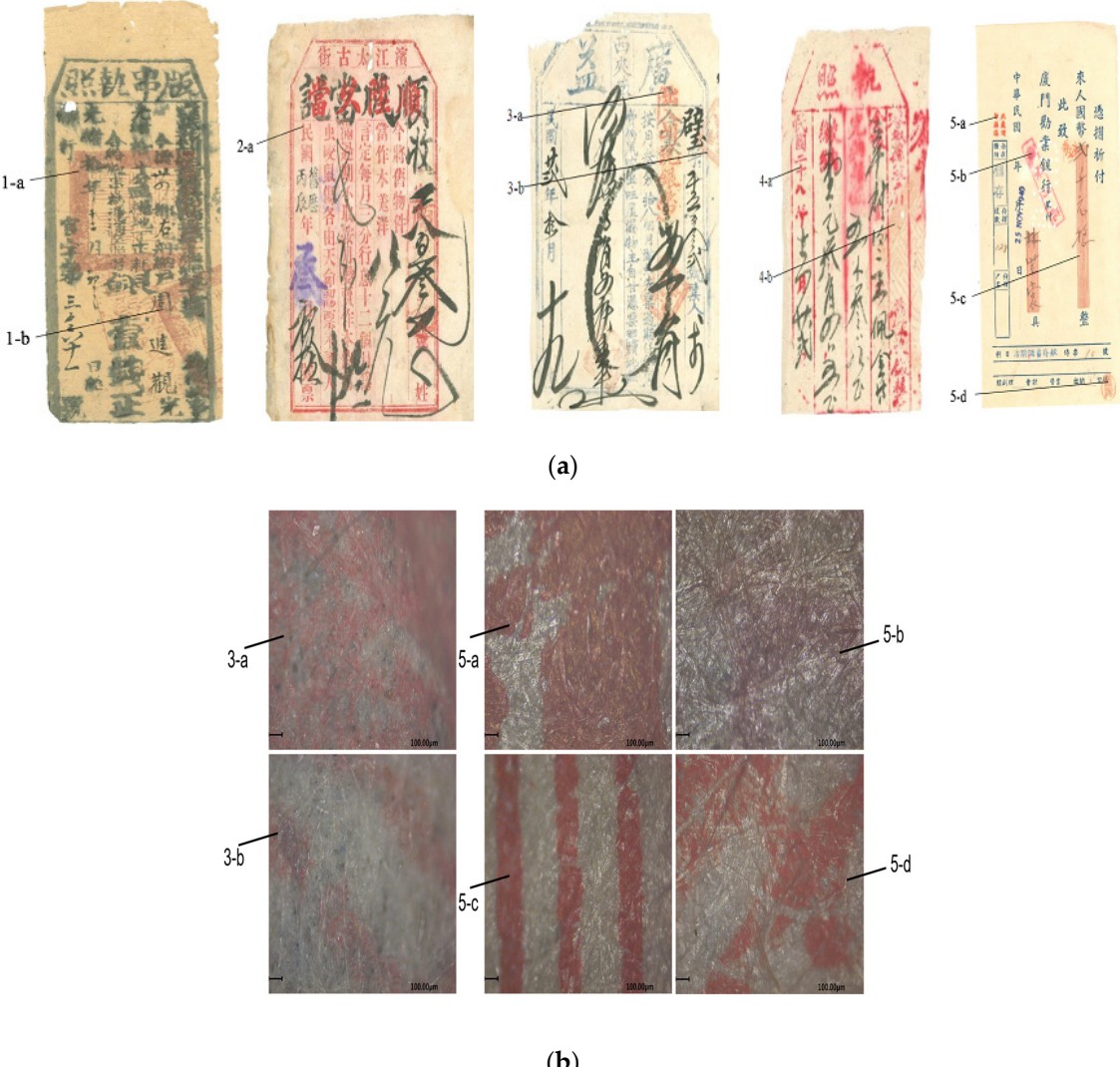

**Figure 1.** (**a**) Historical Chinese stamps investigated in this study. Selected areas are marked in the figure with small (blue) circles (the dates, from left to right are 1884, 1916, 1933, 1939, 1940). (**b**) The optical microscopy images, taking the points on the pictures of sample 3 and sample 5 as examples.

Red marks are noticeable on all five notes. Small dots (marked by small circles) were chosen on each paper note for optical measurements, and each was referred to as "a" or "b" (for example, 1-a, and 1-b represent dots "a" and "b" on Sample 1, respectively). Sample

1 is a bank bill from the 1884 issue, of 23.4 cm × 9 cm size. The body adopted the blue pigment printing, and the center was covered by a red square seal, as shown in area 1-a; the lower right corner of the sample was stamped by a red cross-page seal, as shown in area 1-b. Sample 2 is a 15 cm × 12 cm bill issued in 1916. The bill body adopted red pigment printing and was stamped with purple and black handwriting, as shown in sample 2-a. Sample 3, of 8.8 cm × 13 cm size, is a bill issued in 1933. The bill body adopted blue pigment printing, and the right side of the body was stamped with a red rectangular seal, "In and/or out of the Current Silver Coin", in area 3-a; the right edge was stamped with a diamond "Guang Yi" red seal in area 3-b. Sample 4, sized at 14.5 cm × 10 cm, is a 1939 issue note. The nominal subject is red in area 4-a, and area 4-b has a red chapter cover on the right side of the seal. Sample 5 is a 1940 print issue note sized at 16.4 cm × 8.8 cm. The nominal subject is blue, the far left in area 5-a is printed with "Do not fill in here" in red letters, the center was stamped by a "transfer paid" rectangular red seal in area 5-b, the central printing of the bill has 2 red vertical stripes in area 5-c, and the lower right corner of a round red seal is affixed in area 5-d.

*2.2. Instrumentation*

A XploRA micro-Raman spectrometer (Horiba Jobin-Yvon, Palaiseau, France), coupled with a BX-51 confocal microscope (Olympus, Tokyo, Japan) was utilized for Raman analysis. All tests were carried out in a dark room (room temperature: 25 °C), and the Raman spectrometer's working temperature was reduced to −70 °C. All samples were examined at a 100× microscope objective to locate the areas for Raman analysis. The laser spot size was approximately 1 μm, and the laser power reached approximately 2~3 mW on the samples. The conditions for recording each Raman spectrum are as follows: 50 s exposure time, 2 times of accumulation, and 100~3500 cm$^{-1}$ spectral range.

A JEOL JSM-6610LA SEM-EDS (JEOL, Peabody, MA, USA) was employed to distinguish elemental content in detail with a secondary electron (SEI) signal, magnification by 50~100; the EDS conditions included a 20 kV accelerating voltage, a 10 mm working distance (WD), and an objective lens diaphragm of 3, the chemical compositions were determined by a JEOL EX-94300S4L1Q Energy Dispersive Spectrometer (JEOL, Peabody, MA, USA). All measurements were carried out at room temperature. FT-IR (Nicolet 6700 FT-IR, Thermo Fisher, Waltham, MA, USA) measurements were also employed. The test range is from 500 to 4000 cm$^{-1}$. The optical microscope used was a VHX-5000 Confocal Laser instrument (Keyence, Osaka, Japan).

**3. Results**

The XploRA Raman spectrometer was employed to test the red pigments of all the samples. The results are shown in Figure 2. Based on the analyses in Figure 2, the marked red pigments in areas 1-a and 1-b produced Raman signal bands between 600 and 150 cm$^{-1}$. The Raman spectra vibrational bands of area 1-a and area 1-b at 284, 402, and 605 cm$^{-1}$ correspond to symmetric Fe–O bending, and the signals at 218 cm$^{-1}$ and 228 cm$^{-1}$ correspond to symmetric Fe–O stretching [1]. The presence of these peaks indicates that the main component of this pigment is ocher [4–6]. Cinnabar was also applied as a coloring material for ancient articles. The characteristic Raman peaks of area 3-a, 3-b, 4-b, and 5-d showed evidence of mercury sulfide (HgS) pigment, with bands at 252, 284, and 344 cm$^{-1}$ related to the trigonal mercury sulfide structure [12]. Cinnabar's wide application could be due to economic and technological motivations.

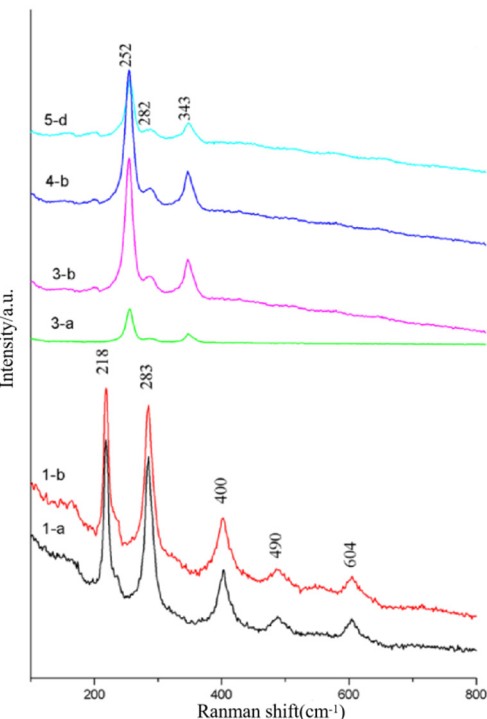

**Figure 2.** Raman spectra of areas 1-a, 1-b, 3-a, 3-b, 4-b, and 5-d.

The characteristic Raman bands were difficult to detect in areas 4-a, 5-a, and 5-b. According to the morphological characteristics, it was speculated that the material might be an organic pigment. Based on this logical inference, FT-IR spectrometry could be a better option to test the marked area pigments. FT-IR absorption peaks of the areas 4-a, 5-a, and 5-b are almost similar, as shown in Figure 3a. These peaks match the spectrum of a $C_{16}H_{12}N_2$ infrared standard in the Spectral Database for Organic Compounds (SDBS) library, and the corresponding structure is shown in Figure 4. Based on the red color attributes of the samples, it can be inferred that the compound of structure $C_{16}H_{12}N_2$ is similar to magenta $C_{18}H_{12}N_2O_6SCa$ [15], as shown in Figure 5. Both of the results have chromophores, auxochrome groups, and conjugated structures. To determine whether the measured pigment is indeed magenta, a standard magenta sample was measured again by FT-IR. Absorption peaks at 1600, 1580, 1500, and 1450 cm$^{-1}$ could correspond to the characteristic vibrations of a benzene ring. The stretch at 1554 cm$^{-1}$ can originate from the C–N stretching vibration. The band at 1340 cm$^{-1}$ can be assigned to the vibration of a C–N group. Compared with the magenta standard sample and the FT-IR spectra of areas 4-a, 5-a, and 5-b in Figure 3a, the characteristic peaks at 3837, 2356, 1874, 1554, 1039, 941, and 848 cm$^{-1}$ are roughly the same. Therefore, the red pigment in areas 4-a, 5-a, and 5-b might be the same type of red organic pigment with a similar structure to magenta. It can be also seen from Figure 3b that the FT-IR spectra of the paper and sample 5-b are consistent, and the sample 5-b is a test point with pigment, and its absorption peak is much stronger than that of paper, especially the absorption peaks are at 3337, 2914, 1493, and 1015 cm$^{-1}$, respectively.

The basic composition of paper is plant fibers, and the auxiliary materials are mainly composed of adhesive, filler, and pigments. Rubber is applied to resist liquid penetration and diffusion. Fillers including talcum powder ($Mg_3[Si_4O_{10}](OH)_2$), calcium carbonate ($CaCO_3$), and kaolin ($Al_2O_3 \cdot 2SiO_2 \cdot 2H_2O$) impart specific properties such as opacity and brightness to the paper. Pigments are mainly used to dye and modulate the color on paper [16].

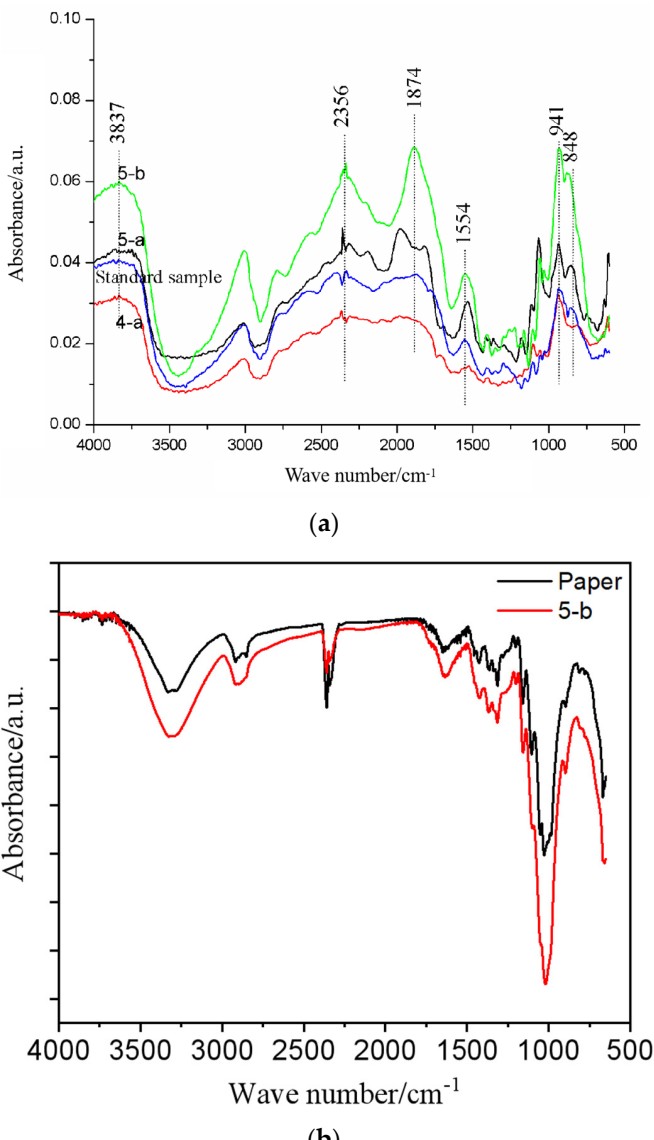

**Figure 3.** (**a**) IR spectrum of areas 4-a, 5-a, and 5-b and the magenta standard sample. (**b**) IR spectrum of areas 5-b and paper.

**Figure 4.** Structure of $C_{16}H_{12}N_2$.

**Figure 5.** Structure of $C_{18}H_{12}N_2O_6SCa$.

No Raman scattering effect was observed in the red pigments of area 2-a in sample 2 and area 5-c in sample 5, separately. SEM-EDS can be used to investigate the elements of the lesser components and thin layers of pigments. Referring to the left color spline, the content is higher in areas that are closer to the color of the upper part. The red pigments of area 2-a in sample 2 were measured by microscopic observation and energy dispersive spectrum mapping analysis, as shown in Figure 6. The paper and ink included Al, S, Si, Ca, Cl, Zn, Ba, O, and Pb. Herein, Si, Ca, and Ba are mainly distributed as filler components in the paper, and the structures can be further inferred as corresponding to the fillers $Al_2O_3 \cdot 2SiO_2 \cdot 2H_2O$, $CaCO_3$, and $BaCO_3$, respectively. The distributions of Al, S, Cl, Zn, Ba, Pb and other elements are consistent with red writing, and the elements are mainly distributed in red ink as pigments or fillers. Pb can be considered the only color element according to the attributes of the red pigment, which is preliminarily inferred as the lead compounds $Pb_3O_4$ or $PbO$. In addition, zinc sulfide ($ZnS$) may also be added for color modulation, which made the sample closer to orange red from the sample appearance.

The red pigments of area 5-c in sample 5 were also measured by energy dispersive spectrum mapping analysis, as shown in Figure 7. The paper and ink included Mg, Al, S, Si, Ca, Cl, Ba, O, and Pb. In general, Mg, Si, Al, and Ba were mainly distributed as filler components in the paper, and the structures can be inferred as $Mg_3[Si_4O_{10}](OH)_2$, $Al_2O_3 \cdot 2SiO_2 \cdot 2H_2O$, and $BaCO_3$. The distributions of S, Ca, Ba, Pb, and other elements are consistent with red writing, and the abovementioned elements are likely mainly distributed in red ink as pigment or fillers. Pb is the only color element, according to the attributes of the red pigment, which is preliminarily inferred to be the lead compounds $Pb_3O_4$ or $PbO$.

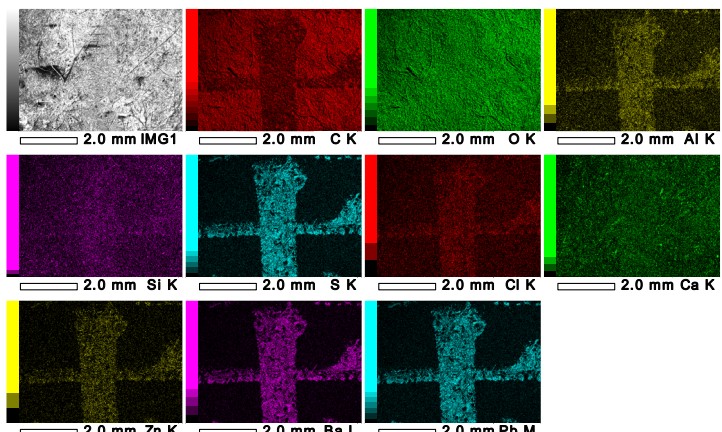

**Figure 6.** SEM-EDS mapping image of area 2-a.

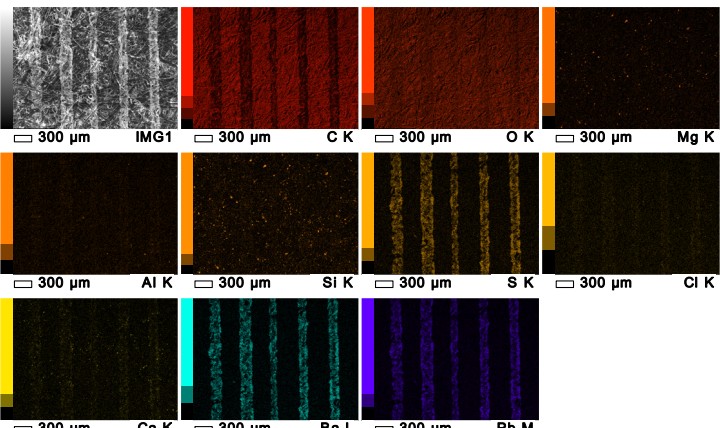

**Figure 7.** SEM-EDS mapping image of area 5-a.

## 4. Conclusions

The red pigments of samples were analyzed by laser confocal Raman spectroscopy, scanning electron microscopy energy dispersive spectroscopy (SEM-EDS), and infrared spectroscopy. The red pigments of area 1-a and area 1-b in sample 1 are ochre, and the red components of areas 3-a and 3-b in sample 3, area 4-b in sample 4, and area 5-d in sample 5 are cinnabar. The orange red pigments in area 2-a in sample 2 indicate that Al, S, Cl, Zn, Ba, Pb and other elements are mainly distributed in red ink. The red pigments in area 5-c in sample 5 show this red ink contains S, Ca, Ba, Pb, and other elements, and the red pigments in area 4-a in sample 4, area 5-a, and area 5-b in sample 5 may be magenta. Based on the detection and analysis of the red pigments on the Qing Dynasty and the Republic of China bill series, the red pigments on early negotiable paper notes are essentially inorganic pigments, while the red pigments of most modern paper notes are organic pigments. Paper printing materials have evolved from inorganic pigments to organic pigments, which reflect the necessity and practicality of anti-counterfeiting and the influence of the development of industrial technology on the printing industry. The three abovementioned combined technologies are a powerful means to analyze bill pigments, which has broad application potential for the analysis of presswork and historic paper relics.

**Author Contributions:** Investigation: J.R., C.G., Y.S., and J.W.; funding acquisition: J.W. and W.C.; writing—original draft preparation: Q.L. and J.S. All authors have read and agreed to the published version of the manuscript.

**Funding:** This research was funded by the National Natural Science Foundation of China Grant Nos. (52002074, 2016YFA0202302, 61527817, 61845236, 52002074, 51602071 and 11427808), Beijing Social Science Foundation No. 15LSC014, Beijing Natural Science Foundation (4132031), Beijing Postdoctoral Research Foundation, Beijing University Student Research Program, Beijing City College High-level Teachers Team Construction Program for the Young Top Talents Training CIT&TCD201904050. The Open Research Subject of Key Laboratory of Dielectric and Electrolyte Functional Material Hebei Province No. HKDE201902.

**Institutional Review Board Statement:** Not applicable.

**Informed Consent Statement:** Not applicable.

**Data Availability Statement:** Data sharing not applicable.

**Conflicts of Interest:** The authors declare no conflict of interest.

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
