# Peer review of "Non-invasive Optical Technical Identification of Red Pigments on Chinese Paper Notes"

_coatings, doi:10.3390/coatings11040410_

Round 1
Reviewer 1 Report
The study is interesting but some of the conclusions are not supported by analytical evidences. In particular, I am surprised that the authors did not identify red lead by means of Raman spectroscopy, which is relatively easy to obtain. They stated that this pigment was present only on the base of SEM-EDS, but this is not correct.
In addition, I suggest thorough revision of the English language because in several instances the text is unclear.
In the end, the work deserves some interest and can be considered for publication on Coatings journal, but it needs major revision. The attached file includes some more corrections to do.

Author Response
We thank the reviewer for the positive assessment of our work and the constructive suggestions, which helped us to further improve the manuscript. We have fully addressed the comments in the revised manuscript. The revised sentences and words are in highlight to make them easy to find in the manuscript. Our point-by-point responses to reviewer comments are itemized below, with our response also in blue fonts.

Reviewer 2 Report
The paper deals with red pigments of traditional paper notes analysis aimed at understanding the chemical composition of the used dyes. Raman spectroscopy coupled with Infrared spectroscopy and SEM-EDS analysis allowed identification of the pigment base (both inorganic and organic) as well as of other additives used for colour modulation or supplier of other specific properties to the paper. The interesting series of the identified inorganic colours ocher, cinnabar, and red lead as well as synthetic magenta was buttressed by the used analytical methods. These include also the presence assessment of specific elements contained in the identified compounds. The paper offers a general hint for a methodological approach in this field and therefore deserves publication in the journal. Nevertheless, since the samples cover a wide timespan ranging from 1884 (sample 1) to 1940 (sample 5) a more in deep discussion of the use of different colours with the time, possibly with eventually available further references could improve the outcome of the manuscript. Moreover, at least the issuing date of each note should be also reported in a possibly more detailed caption to Fig.1 while Fig.3 and Fig.4 could merge in a single Figure and a more detailed caption to the present Fig 6 and Fig. 7 is advisable.
Author Response

(The authors gave the same response as above.)

Reviewer 3 Report
The paper in question deals with the characterization of old document inks typical of Chinese cultural heritage. The authors used different techniques to assess the kind of used pigment for each document; micro-Raman spectroscopy, FT-IR and electron dispersive X-ray spectroscopy were used for the characterization. The Paper is interesting and suitable for publication in «Coatings», even if some revisions are required before publication.
A - English is in general well used and udertandable. The exposition of concepts is subsequential and logic. However, it needs to be improved; for example, in the Abstract (line 20) «added as color modulation.» should be modified in «added as color modulator.» or «added for color modulation.». This is an example, other improvements are also possible.
B - The subject is very interesting, but the analyses are not carried out in a systematic way: substrate analyses are missing, for example, and not all the results related to the presented points are shown. At the same time, it is not explained why they are not present (same results as another point? not significance of the result? etc.). So, even if the paper is interesting, a more methodic characterization would be required.
B - The authors used FT-IR for the analyses of some samples. This technique should be used for all the samples; otherwise, it is difficult to compare things which are not the same. In addition, the parameters used for FT-IR investigations should be put in the Materials and method section.
C - How were the sample prepared for electron microscopy? In fact, the substrate is composed by cellulose fibers, which are not conductive. Please, explain this
D - The authors should present data related to the substrates, as well, to exclude any peak superposition.
E - An image (even obtained with optical microscopy) of the investigated zone(s) would be appreciated.
F - Figures 2 and 3 quality should be improved: Figure 2 text should be made bigger, while Figure 3 curves should be more separated.
G - «To determine whether the measured pigment is magenta, a standard magenta sample was measured again by FT-IR.» What is a "standard magenta"? Is it pertinent to assume that in the past they used the same type of magenta (organic, synthetic)? How that was produced? are there other possibilities to take into account?
Author Response

(The authors gave the same response as above.)

Reviewer 4 Report
The manuscript focuses on the analysis of red pigments on several Chinese stamps dated between the 19th to the 20th century by means of FTIR, Raman and SEM-EDS techniques. The paper is not innovative either in terms of the studied subject or in terms of analytical methodology. Thus, my first remark would be that the manuscript does not qualify for a short communication as it doesn’t report new important findings. Moreover, in the current form the manuscript has several weakness that must be addressed. The language and style of the paper must be significantly improved as it makes reading difficult and causes misunderstandings. The aim of the study is not clear nor the gap in the literature that the paper wants to address. All sections of the paper allow for improvements (see some specific comments below).
However, the study could be of interest to the scientific community within the cultural heritage field if a wider collection of historical Chinese stamps would be investigated; if possible, the use of imagistic techniques such as multi- or hyperspectral imaging would also add a new dimension to the study. The authors should therefore consider a major revision and, if considerably improvements are made, submit a new manuscript as an original research paper.
Specific comments:
Title: The title should be more specific. Avoid using generic terms such as “optical technical identification”. “Chines paper notes” is unclear; go for “historical Chinese stamps”.
Abstract: References to specific investigated samples have to be avoided (see Lines 16-17). The abstract should give a clear summary of the work without the need for the reader to consult the whole paper.
Introduction: Besides providing a short overview of previous literature in the field, the introduction should also clearly describe the main aim of the study. Reference should be made only to works closely related to the subject under investigation.
Lines 32-37: Information provided here are in my opinion out of context as they refer to studies carried on artifacts from outside China. The main idea of the study is, as previously stated by the authors, the occurrence of red pigments on paper support, in China. If, on the other hand, the study would address the occurrence of red pigments on historical stamps in general, then such references would fit.
Experimental section:
Lines 47-64: A lot of somehow repetitive information is provided on the samples. This section can be more concise. Sample locations can be seen in Figure 1, while information regarding the size of the artifacts could be included in the caption. Focus on the importance of the artifacts/ on the unique characteristics.
Lines 67-78: Were the investigations carried in-situ, directly on the investigated object, or on samples? It is not clear. This aspect should be clarified. The title says non-invasive… do the selected equipment allow analysis of large specimens? Clarify the reader; provide all the important experimental aspects. Details of the FTIR measurements are completely missing. Why?
Results: Discussion of the results should be more structured. Data could be presented in subsections; each technique separately, or in groups of pigments. Be very clear and concise and avoid general discussions (see Lines 114-124). Speculations should also be avoided - see for example Line 138 (chemical formula of inferred fillers).
Language and style: Particular attention to technical terms and scientific language in general should be given. See below some inappropriate examples:
Line 66: "Paper notes and their dots chosen for analyses". How about: Historical Chinese stamps investigated in this study. Selected areas are marked in the figure with small (blue) circles.
Line 110: "the characteristic peaks at 3837... correspond to gaps between the absorption bands". What do you mean?
Line 115: "composed of rubber, filler, and pigments". I believe by rubber you mean adhesive, or glue?
Line 145: "energy spectrum analysis of mapping". EDS mapping analysis.
Author Response

(The authors gave the same response as above.)

Round 2
Reviewer 1 Report
I consider all the issues raised in the previous stage addressed by the authors. Therefore the work can be published in the present form.
Author Response
We thank the reviewer very much for the comments.
Reviewer 3 Report
English needs to be improved. The paper can be published.
Reviewer 4 Report
The authors apparently tried to address some of my comments but the manuscript has not significantly improved. Some minor changes of the text are not by far enough to qualify this as major revision. The paper is still badly written both in terms of style and analytic discussion. There are several analytical aspect that were incorrectly addressed. As stated in the first round of review I recommend to the authors to expand and fully rewrite their study.